# Preparation, Characterization, and Antioxidant Properties of Selenium-Enriched Tea Peptides

**DOI:** 10.3390/foods12224105

**Published:** 2023-11-12

**Authors:** Kang Wei, Yang Wei, Peng Zhou, Jiangxiong Zhu, Lanlan Peng, Lizeng Cheng, Yuanfeng Wang, Xinlin Wei

**Affiliations:** 1School of Agriculture and Biology, Shanghai Jiao Tong University, 800 Dongchuan Road, Shanghai 200240, China; wk5516828@sjtu.edu.cn (K.W.); sunny1994@sjtu.edu.cn (Y.W.); zjx261023@163.com (J.Z.); pengll@sjtu.edu.cn (L.P.); dr_cheng@sjtu.edu.cn (L.C.); 2College of Life Science, Shanghai Normal University, 100 Guilin Road, Shanghai 200234, China; a461802294@163.com

**Keywords:** selenium, tea protein, tea peptide, identification, antioxidant activity

## Abstract

The research on the activity of selenium (Se)-enriched agricultural products is receiving increasing attention since Se was recognized for its antioxidant activities and for its enhancement of immunity in trace elements. In this study, antioxidant Se-containing peptides, namely, Se-TAPepI-1 and Se-TAPepI-2, were optimally separated and prepared from Se-enriched tea protein hydrolysates by ultrafiltration and Sephadex G-25 purification, and subsequently, their physicochemical properties, oligopeptide sequence, and potential antioxidant mechanism were analyzed. Through the optimization of enzymatic hydrolysis conditions, the Se-enriched tea protein hydrolyzed by papain exhibited a better free radical scavenging activity. After separation and purification of hydrolysates, the two peptide fractions obtained showed significant differences in selenium content, amino acid composition, apparent morphology, peptide sequence, and free radical scavenging activity. Therein, two peptides from Se-TAPepI-1 included LPMFG (563.27 Da) and YPQSFIR (909.47 Da), and three peptides from Se-TAPepI-2 included GVNVPYK (775.42 Da), KGGPGG (552.24 Da), and GDEPPIVK (853.45 Da). Se-TAPepI-1 and Se-TAPepI-2 could ameliorate the cell peroxidation damage and inflammation by regulating NRF2/ARE pathway expression. Comparably, Se-TAPepI-1 showed a better regulatory effect than Se-TAPepI-2 due to their higher Se content, typical amino acid composition and sequence, higher surface roughness, and a looser arrangement in their apparent morphology. These results expanded the functional activities of tea peptide and provided the theoretical basis for the development of Se-containing peptides from Se-enriched tea as a potential natural source of antioxidant dietary supplements.

## 1. Introduction

Tea (*Camellia sinensis* L.) is one of the most ancient and popular beverages worldwide because of its unique aroma and taste [1,2]. Tea has emerged as the second-largest beverage globally with consumption reaching 7.4 million tons [3]. Tea has shown a great potential to lower the risk of a wide range of diseases, such as cancer, metabolic disorders, immune diseases, mental disorders, and neurodegenerative diseases [4,5,6,7]. Tea contains rich natural active components, including tea polyphenols, theanine, tea polysaccharides, tea proteins, and other nutrients, among which polyphenols and polysaccharides may be the main components to exert the activity of tea. According to previous reports, tea polyphenols and tea polysaccharides exhibit strong anticancer and antioxidant activities [8,9]. In our earlier research, dark-tea polysaccharides showed a high antioxidant activity [10]. Moreover, theanine and theabrownin have also shown a respective biological activity [11,12]. Nevertheless, to the best our knowledge, the biological activity of tea protein and its hydrolysates has rarely been reported.

Generally, tea contains 21–28% of protein, the majority of which are water-insoluble, with the exception of globulin [13]. Traditional tea preparation involves boiling water or a decoction, which causes a significant loss of water-insoluble protein [14]. Crude protein can be employed in preparing polypeptide by hydrolysis, which has a great potential for various biomedical applications [15]. Peptides derived from plant protein usually exhibit favorable antioxidant activities, such as rice bran peptides, corn antioxidant peptides, and soybean antioxidant peptides [16,17,18].

Selenium (Se), as an essential trace element in the human body, is closely related to human health [19]. It was reported that Se-containing peptides from tea protein hydrolysates showed a high antihypertensive activity [15]. In particular, due to the essential component of glutathione, Se is crucial to the antioxidant activity of the body [20]. Se-enriched peptides from brown-rice protein was reported to be used as a natural food-born antioxidant, which was attributed to the antioxidant activity of Se [21]. Notably, it was reported that compared to regular green tea, selenium-enriched green tea exhibited a higher antioxidant activity [21], which might be related to the Se-rich tea protein. Nevertheless, there has been little discussion about the preparation and purification of Se-enriched tea protein and peptides. The antioxidant bioactivity of Se-enriched tea protein and peptides are still unclear [14]. Correspondingly, further research is needed to understand the structure–function relationship of Se-enriched tea peptides.

In this study, the preparation process of selenium-containing peptides from tea was optimized. The Se-containing peptide fraction with the highest antioxidant activity was screened, and its physicochemical characterization was identified. Furthermore, the potential mechanism of antioxidant activity of Se-containing peptides was investigated. Our research will provide a new scheme for the high-value utilization of tea protein.

## 2. Materials and Methods

### 2.1. Materials and Reagents

Se-enriched green tea leaves were obtained from Yichang, Hubei Province, China. Human liver cell lines (LO2) were obtained from BeNa Culture Collection (Xinyang, China). The Se standard solution (100 mg/L) and Sephadex G-25 were purchased from the Shanghai Institute of Measurement and Testing Technology (Shanghai, China) and GE Co. (Pittsburgh, PA, USA), respectively. 2,2′-azino-bis (3-ethylbenzothiazoline-6-sulfonic acid) (ABTS•), 1,1-diphenyl-2-picrylhydrazyl (DPPH•), alcalase (500 U/mg), neutrase (500 U/mg), papain (500–800 U/mg), and trypsin (>2500 U/mg) were purchased from Aladdin Reagent Co., Ltd. (Shanghai, China). Superoxide dismutase (SOD), glutathione peroxidase (GSH-Px) and catalase (CAT) kits were purchased from the Jiancheng Bioengineering Institute (Nanjing, China). Antibodies of nuclear factor (erythroid-derived 2)-like 2 (Nrf2), heme oxygenase-1 (HO-1) were obtained from Santa Cruz Co. (Dallas, TX, USA), antibodies of NAD(P)H quinone oxidoreductase-1 (NOQ-1) and glutamate-cysteine ligase catalytic subunit (GCLC) were obtained from Abcam Co., Ltd. (Cambridge, UK). All other chemicals and solvents were of analytical grade unless otherwise specified.

### 2.2. Preparation of Se-Enriched Alkali-Soluble Tea Proteins

The Se-enriched green tea was smashed and extracted twice by 60% ethanol with a ratio of 1:10 (*w*/*v*) at 40 °C for 1 h, and then centrifuged at 5000 rpm for 10 min. Next, the residues after centrifugation were added into boiling water with a ratio of 1:10 (*w*/*v*) for 2 h and then centrifuged at 5000 rpm for 10 min. The supernatant was discarded to remove water-soluble substances, such as tea polysaccharides. Subsequently, a single-factor optimization for the protein extraction conditions was performed, such as the concentration of alkali (NaOH solution with concentration of 0.01, 0.05, 0.10, and 0.20 M) and cross-linked polyvinylpyrrolidone (PVPP, 2, 4, 6, and 8 mg/mL), as well as temperature (20, 30, 50, and 70 °C). Briefly, we added a mixture of alkali and polyvinylpyrrolidone at a solid–liquid ratio of 1:40 to the tea residue for a 2 h extraction under an optimal temperature, followed by a centrifugation (Anting LXJ-IIB, Shanghai, China) at 5000 rpm for 10 min. Thereafter, the precipitation was extracted once using the same method, and the supernatant was precipitated with an equal volume of acetone, then centrifuged at 3000 rpm for 10 min. The precipitated protein was added to the extraction solution and precipitated with acetone triplicate. Finally, the protein precipitate was added to ultrapure water, dialyzed through a dialysis bag (3500 Da) for 3 days, and freeze-dried to obtain Se-enriched tea alkali-soluble proteins (Se-TAPs).

### 2.3. Single-Factor and Response Surface Methodologies (RSM) for Alkali-Soluble Se-Enriched Tea Alkali-Soluble Peptides (Se-TAPeps)

The Se-TAP was hydrolyzed by four different proteases (including alcalase, neutrase, papain, and trypsin), and the condition of enzymatic hydrolysis is presented in Appendix A. In short, 3% (*w*/*v*) Se-TAP was mixed with a certain amount of hydrolase under the optimum pH and temperature and then was hydrolyzed by different hydrolases for 4 h. Next, the hydrolases were terminated by boiling for 10 min. The Se-TAP hydrolysates were separated by centrifugation at 5000 rpm for 15 min for a further assessment of the antioxidant capacity, including DPPH•, ABTS•, hydroxyl radical (•OH), and superoxide anion radical (O_2_^−^) scavenging. The enzymatic hydrolysis conditions were optimized by a single-factor experiment, including pH (4, 5, 6, 7, 8, and 9), temperature (40, 50, 60, 70, and 80 °C), enzyme concentration (500, 1500, 2500, and 3500 U/g), and duration (1, 2, 3, 4, 5, and 6 h), which set the •OH scavenging capability as the evaluation index. Based on single-factor experiments, the response surface methodology was employed to explore the implications of the pH, temperature, enzyme concentration, and duration on the •OH scavenging capacity of protein hydrolysate (Design-Expert 7.0.0 Trial, State-Ease, Inc., Minneapolis, MN, USA). Following the Central Composite Design (CCD) approach, we employed four independent variables with five different levels and conducted a total of 30 experiments. The specific factors and their corresponding levels can be found in Appendix A. The antioxidant experiments including DPPH•, ABTS•, •OH, and O_2_^−^ scavenging activities were performed as in the following, and the sample concentration was set as 1000 μg/mL.

### 2.4. Detection of Antioxidant Activity and Selenium Values

DPPH• scavenging: The DPPH• radical scavenging activity was measured according to the method described by a previous report with a slight modification [10]. In short, a sample solution (1 mL) was mixed with the DPPH–alcohol solution (1 mL, 0.1 mM) as a sample group. Meanwhile, DPPH• (1 mL, 0.2 mM) was used as a blank, and DPPH• (1 mL, 0.2 mM) mixed with ethanol (1 mL, 95%) was used as the control group. The samples were stored at room temperature for 30 min in the dark, and the absorbance were determined at 517 nm. Vitamin C (Vc) was selected as a positive control. The formula of the DPPH• radical scavenging activity is as follows:DPPH• scavenging rate (%) = [1 − (A0 − A1)/(A2 − A1)] × 100 (1)
where A0, A1, and A2 represent the absorbance of the sample group, blank group, and control group, respectively.

ABTS• scavenging: the ABTS• scavenging activity was evaluated according to our previous method [10].

•OH scavenging: The scavenging capacity of the sample solution on •OH radicals was evaluated by a modified reported method [10]. Briefly, FeSO_4_ (1 mL, 9 mM), H_2_O_2_ (1 mL, 8.8 mM), and a salicylic acid–ethanol solution (1 mL, 9 mM) were added to 1 mL of peptide solution with certain concentrations and incubated at 37 °C for 60 min. The absorbance of the sample was recorded at a wavelength of 510 nm, and Vc was selected as a positive control. The scavenging activity was calculated using the following equation:•OH scavenging rate (%) = [1 − (A1 − A2)/A0] × 100 (2)

A0 represents the absorbance reading of the control group without any sample, A1 corresponds to the background absorbance in the absence of hydrogen peroxide, and A2 signifies the absorbance recorded for the sample group.

O_2_^−^ scavenging: the O_2_^−^ scavenging activity was evaluated according to our previous method [10].

The selenium content of the samples in this study was determined using the previous method [22].

### 2.5. Isolation and Purification of Se-TAPeps

The mixed hydrolysate of Se-TPeps after enzymolysis (0.1 g/mL) was ultrafiltered sequentially through 100, 10, and 1 kDa ultrafiltration membranes (Millipore, MA, USA), and then four fractions, including Se-TAPepI (<1 kDa), Se-TAPepII (1~10 kDa), Se-TAPepIII (10~100 kDa), and Se-TAPepIV (>100 kDa) were obtained. After concentration, dialysis, and lyophilization, these four fractions were stored at 4 °C for the evaluation of DPPH• and •OH scavenging activities. Through the evaluation of antioxidant capacity and ultraviolet and visible (UV–vis) spectra at 200–500 nm, the fraction (50 mg/mL) with the highest free radical scavenging activity was passed through a Sephadex G-25 gel filtration column (1.0 × 100 cm) using a flow rate of 1 mL/min, and the eluted fraction was analyzed at 218 nm. Finally, the eluted fractions were lyophilized and stored at 4 °C, with their DPPH• and •OH scavenging activities assessed. The Se content was determined according to our previous study [15].

### 2.6. Fourier Transform Infrared (FT-IR) Spectra Analysis

The FT-IR detection was performed according to a previous study with modification [23]. A total of 2 mg of purified Se-TAPeps was mixed with 200 mg of KBr (potassium bromide), ground uniformly, and then pressed into uniform tablets. After that, the uniform tablet was scanned by an Avatar-360 FT-IR spectrometer (Thermo Nicolet, Waltham, MA, USA) in the range of 400–4000 cm^−1^.

### 2.7. Scanning Electron Microscope (SEM) Analysis

The purified Se-TAPeps were subjected to gold coating in a vacuum using an E-1045 ion sputter coater (Ted Pella, Redding, CA, USA) [24]. The images could be observed and recorded (10,000×) by a HITACHSU8010 scanning electron microscope (JEOL, Tokyo, Japan).

### 2.8. Amino Acid Composition Analysis

The amino acid composition (sixteen kinds of amino acids) analysis was performed by the L-8900 amino acid analyzer (Hitachi, Tokyo, Japan) according to our previous study [15].

### 2.9. Peptide Sequence Analysis

An ultraperformance liquid chromatography (UPLC) system coupled to a quadrupole–time-of-flight mass (Q-TOF-MS) spectrometer (Waters, Framingham, MA, USA) was used to analyze the peptide sequence. Briefly, the desalted sample underwent separation using a gradient of 80% acetonitrile with 0.1% formic acid, the UPLC test was performed according to a previous report [25], and the MS/MS data were analyzed by Peaks Studio (Bioinformatics Solutions Inc., Waterloo, ON, Canada).

### 2.10. Cell Culture and Treatment

LO2 cells were cultured in DMEM complete medium containing 10% fetal bovine serum (FBS) (Gibco, Waltham, MD, USA) [26]. In addition, cells were incubated in a constant environment (37 °C, 5% CO_2_).

#### 2.10.1. MTT Assay

An MTT analysis was performed using the method of Wen et al. with a slight modification [27]. LO2 cells were seeded in a 96-well plate (1 × 10^4^ cells/well) and cultured with DMEM for 2 h. Subsequently, a series of concentration gradients of TPepI-1 (50, 100, 200, 500, and 1000 μg/mL) were added and incubated for 24 h. Next, 200 mM H_2_O_2_ was added and incubated for 4 h. The untreated group was regarded as the model control (MC). Thereafter, 20 μL of 5 mg/mL thiazolyl blue (MTT) was added in the MC and incubated for 4 h, and then the original medium was removed. Finally, 100 μL of dimethyl sulfoxide (DMSO) was added and shaken for 20 min, and then it was measured at 490 nm. The cell viability was calculated as the ratio of absorbance to normal control cells without sample treatment.

#### 2.10.2. Reactive Oxygen Species (ROS) and Antioxidant Enzyme Detection

ROS detection was performed by a modified reported method [27]. LO2 cells were seeded in a 6-well plate (10^5^ cells/well) and incubated with 500 μg/mL Se-TAPepIs for 24 h. Next, 1 mL of 2 mM H_2_O_2_ solution was added and incubated for 4 h. The cells were divided into four groups, including a model control, normal control, model control + Se-TAPepI-1, and a model control + Se-TAPepI-2. The expression level of ROS in each group was observed and recorded using fluorescence inverted microscopy (Olympus, Tokyo, Japen), and a quantitative analysis was carried out using a reagent kit (Nanjing Jiancheng Bioengineering Institute, Nanjing, China). The enzyme activities of SOD, GSH-Px, and CAT were detected by the kit, respectively.

#### 2.10.3. RT-qPCR Assay

The total RNAs of cells were extracted with TRIzol Reagent (Invitrogen, Carlsbad, CA, USA). The mRNA expression was detected by SYBR qPCR Master Mix (High ROX, Wuhan, China). The used primers are presented below: *NRF2*, F-CCAGCACATCCAGTCAGAAACC, R-GTAGCCGAAGAAACCTCATTGTC; *HO-1*, F-AGGAGGTCATCCCCTACACA, R-GGGCAATCTTTTTGAGCACCTG; *IL-1β*, F-TGAACTGAAAGCTCTCCACCT, R-TTGGGATCTACACTCTCCAGC; *TNF-α*, F-TCTACTCCCAGGTCCTCTTCAAG, R-GGAAGACCCCTCCCAGATAGA; *GAPDH*, F-GGAAGCTTGTCATCAATGGAAATC, R-TGATGACCCTTTTGGCTCCC. The employed internal reference gene was *GAPDH*, and the mRNA expression of the target gene was then measured with the 2^−ΔΔCt^ method.

#### 2.10.4. Western Blot Assay

The cells were lysed with the RIPA lysate (Solarbio, Beijing, China). Total proteins were collected, and their concentration was calculated. A Western blot was performed according to a previous study [26]. Briefly, a 40 μg protein/lane separation was performed by SDS-PAGE, and the protein was transferred with a PVDF membrane. In addition to blocking the protein for 1 h with 5% skimmed milk, the primary antibodies, including NRF2 (*v*/*v* 1:500), and the HO-1 (*v*/*v*, 1:500), NQO-1(*v*/*v*, 1:500), and GCLC (*v*/*v*, 1:500) polyclonal antibodies were mixed with the target protein for 2 h at room temperature. The protein was washed and then incubated with HRP at 37 °C for 1 h. The relative protein quantification was achieved by comparing the grayscale values of the lanes in each sample.

### 2.11. Statistical Analysis

A mean ± SD was calculated for each measurement. Data were analyzed by using SPSS 22.0 (IBM, Armonk, NY, USA) and GraphPad Prism 9.0 (La Jolla, CA, USA). Two groups were compared using the student’s *t*-test with *p* < 0.05 regarded as statistically significant.

## 3. Results and Discussion

### 3.1. Free Radical Scavenging Activity of Se-Enriched Alkali-Soluble Tea Protein Hydrolysate

The free radical scavenging activities of the Se-TAP hydrolysate after enzymatic hydrolysis by four proteases (including alcalase, neutrasa, papain, and trypsin) are shown in Figure 1a–d. The scavenging activities of the hydrolysates after papain enzymatic hydrolysis against the four free radicals (DPPH•, ABTS•, •OH, and O_2_^−^) were the strongest (93%, 99.8%, 57.5%, and 78.7%, respectively), while the free radical scavenging activities of Se-TAPs without enzymatic hydrolysis were the weakest (23.3%, 98.8%, 47.6%, and 51.5%, respectively), indicating that the enzymatic hydrolysis could significantly (*p* < 0.05) enhance the free radical scavenging activity of tea alkali-soluble protein. A possible explanation might be that the enzymatic hydrolysis of tea protein produced small-molecular peptides with a higher scavenging activity [17]. The different enzymes correspond to distinct protease cleavage sites, resulting in varying peptide molecular weights, thus leading to differing antioxidant properties and free radical elimination rates. Additionally, various hydrolysates showed the strongest scavenging performance for ABTS• and the weakest scavenging performance for •OH. Notably, despite the fact that each hydrolysate showed a high free radical scavenging activity, it still had a modest gap with the positive control group except for the ABTS• scavenging activity. The Se-TAP hydrolysate produced through the papain catalysis exhibited the highest antioxidant activity when compared to the hydrolysates from other enzymes, consistent with the findings of Zarei et al., who found that the peptides prepared with papain contained more hydrophobic residues with a basic and neutral isoelectric point, exhibiting a higher radical scavenging activity [28]. Therefore, papain was chosen for the subsequent preparation of antioxidant tea peptides.

### 3.2. Model Fitting

It was evident that RSM were great to analyze and optimize the preparation process of antioxidant peptides [29]. Zhuang et al. used RSM to obtain an optimum hydrolysis condition for the antioxidant peptides from tilapia skin with the strongest •OH scavenging ability. In this study, a single-factor test was performed to explore the impact of different test factors on the •OH scavenging rate (Figure 1e–h). The CCD model was used to generate 30 groups of tests based on the findings of the single-factor experiment (Appendix A). The •OH scavenging rate exhibited a response range from 47.82% to 66.30%, indicating that various factors and their levels had a substantial influence on antioxidant activity. The relationship between •OH scavenging and the experimental variables could be described using the quadratic regression equation below:Y = −369.24 + 34.6X_1_ + 8.22X_2_ + 0.04X_3_ + 27.84X_4_ + 0.03X_1_X_2_ + 0.0018X_1_X_3_ − 0.85X_1_X_4_ − 0.00069X_2_X_3_ − 0.13X_2_X_4_ + 0.00085X_3_X_4_ − 2.47X_1_^2^ − 0.06X_2_^2^ − 0.0000038X_3_^2^ − 2.36X_4_^2^(3)
where Y is the •OH scavenging (%), and X_1_, X_2_, X_3_, and X_4_ are the coding variables of pH, hydrolysis temperature, enzyme concentration, and hydrolysis time, respectively.

According to the analysis of variance (ANOVA) results presented in Appendix A, the quadratic regression model exhibited a *p*-value of 0.0009, signifying its adequacy in explaining the data. Additionally, the R-squared value of 0.8423 indicated that the model could reliably account for 84.23% of the variation. When assessing the influence of different factors, it was observed that X_1_, X_2_X_3_, X_1_^2^, X_2_^2^, X_3_^2^, and X_4_^2^ had statistically significant effects on •OH scavenging. Furthermore, the primary parameters influencing •OH scavenging were ordered as X_1_ > X_3_ > X_2_ > X_4_, i.e., pH > enzyme concentration > temperature > time according to the F values in Appendix A. The three-dimensional response surface may be used to indicate the optimal level of two variables because it can visually illustrate their interactions (Figure 1i–n). As can be seen, the pairwise interactions between the extraction factors were strong (the contours were all oval), especially the interaction between temperature and enzyme concentration (Figure 1l). Moreover, as depicted in Figure 1l, it was evident that the extraction duration had the most pronounced impact on •OH scavenging, aligning with the findings from the ANOVA results. A binomial regression model was employed for the prediction and analysis, leading to the identification of the following optimal process conditions for the Se-TAP hydrolysis: pH = 7.45, 55 °C of hydrolysis temperature, 1900 U/g protein of enzyme dose, and 3.4 h of hydrolysis time. The •OH scavenging rate was obtained (66.73 ± 0.40%) by repeating the experiment three times according to the optimal circumstances established by the analysis and design.

### 3.3. Separation, Purification, and Antioxidant Evaluation of Se-TAPeps

The Se-TAP hydrolyzed by papain was separated by ultrafiltration into three components, namely, Se-TAPepI (<1 kDa), Se-TAPepII (1~10 kDa), and Se-TAPepIII (10~100 kDa), with Se contents of 1.7, 8.5, and 12.7 mg/kg, respectively, indicating that Se was primarily concentrated in the higher-molecular-weight (MW) peptide mixtures. DPPH• is a relatively stable free radical commonly employed for evaluating the antioxidant potential of substances. •OH is the most toxic free radical produced during the human peroxidation process, and it can react with almost all organic biomolecules found in the organism, thus causing irreversible harm to the body [30,31]. Therefore, the scavenging activities of DPPH• and •OH were used to evaluate the antioxidant capacity of Se-TAPeps. It was obvious that increasing the concentration of Se-TAPeps resulted in a more significant enhancement of its radical scavenging activity. The SC50 values (half-maximal scavenging concentrations) of Se-TAPepI, Se-TAPepII, and Se-TAPepIII for scavenging DPPH• were 30.90, 51.76, and 31.41 μg/mL, respectively, while the SC50 values for scavenging •OH were 0.85, 9.43, and 5.38 mg/mL, respectively. Se-TAPepI, characterized by the lowest molecular weight (<1 kDa), displayed the most potent radical scavenging activity among the various fractions. These results align with prior research, where it was observed that low-molecular-weight components (MW < 1 kDa) from bovine casein and egg white hydrolysates exhibited a superior antioxidant activity when compared to other fractions [32,33]. Interestingly, the free radical scavenging capability of three Se-TAPeps was not associated with their selenium content. Studies have indicated that the antioxidant activity of selenium-containing peptides is significantly influenced by the selenium content [34]. Our findings suggested that the effect of MW on the free radical scavenging rate of the Se-TAP hydrolysate might outweigh the influence of Se. Therefore, Se-TAPepI was collected for a further purification analysis.

UV–vis spectra revealed that Se-TAPepI exhibited strong absorption peaks at 218 nm and 274 nm (Figure 2c). The existence of peptide bonds corresponded to the signal at 218 nm, while benzene ring amino acid residues could be indicated by the signal at 274 nm [15,34]. Therefore, the detection wavelength for the Se-TAPepI purification was chosen to be 218 nm. Se-TAPepI was eluted by a Sephadex G-25 gel filtration column to obtain two fractions, Se-TAPepI-1 and Se-TAPepI-2 (Figure 2d), and Se-TAPepI-1 was the major subfraction. The Se contents of these two peptides were 4.03 and 0.59 mg/kg, respectively. The SC50 values of Se-TAPepI-1 and Se-TAPepI-2 for the DPPH• scavenging rate were 18.66 and 46.77 μg/mL, respectively (Figure 2e), and their IC50 values for the •OH scavenging rate were 0.68 and 0.95 mg/mL, respectively (Figure 2f). Therefore, Se-TAPepI-1 had a stronger antioxidant activity compared with Se-TAPepI-2, further implying that the inclusion of selenium may augment the antioxidant potential of tea peptides. Similarly, our previous report revealed that Se-rich tea peptides had a higher hypotensive effect than ordinary tea peptides, indicating that the presence of organic Se can greatly boost the bioactivity of tea peptides [15]. The lower IC50 values revealed that Se-TAPepI-1 had a greater antioxidant activity than many other peptides compared to previously reported antioxidant peptide fractions from rice residue protein, whey protein, and yogurt [35,36,37]. It is worth noting that the selenium content in Se-TAPepI-1 and 2 was 2.4 and 0.8 mg/kg, respectively, revealing that for small-molecule peptides like these, the selenium content is directly proportional to the antioxidant activity.

### 3.4. Structural Characterization of Se-TAPeps

#### 3.4.1. FT-IR Analysis

The FT-IR spectra of Se-TAPepI-1 was depicted in Figure 3a ranging from 4000 to 500 cm^−1^. It was obvious that the absorption peak appearing at 3361.8 cm^−1^ indicated the extension of the N-H and O-H bonds. The absorption at 2933.2 cm^−1^ was attributed to an asymmetric stretch of -CH_3_. Furthermore, the pronounced absorption within the 1500–1690 cm^−1^ range was attributed to the presence of C=C, C=N and N=O bonds. Of particular significance were the absorption peaks observed around 1080 cm^−1^ and 620 cm^−1^, which were linked to the extension of the O-Se-O and Se-O-C bonds [14,38]. These absorption peaks signified the successful modification of the organ selenium group.

#### 3.4.2. SEM Analysis

Figure 3b clearly shows the apparent morphology of Se-TAPeps. Numerous irregularly distributed cross-linking protrusions had the ability to create a network like structure on the surface of Se-TAPepIs, which promoted the intermolecular attractions. Nevertheless, compared to Se-TAPepI-2, Se-TAPepI-1 exhibited a higher surface roughness, looser arrangement, and more visible distribution of loose pores, which may be explained by the fact that the surface of Se-TAPepI-1 appeared to be distributed according to selenization [39].

#### 3.4.3. Amino Acid Composition Analysis

The function of bioactive peptides was significantly influenced by their amino acid composition [40,41]. Table 1 compares the amino acid content distribution of Se-TAPepI-1 and Se-TAPepI-2. In terms of amino acid types, Se-TAPepI-1 shows a more abundant amino acid distribution. Additionally, Se-TAPepI-1 and Se-TAPepI-2 had the highest Gly content, at 28.65% and 12.31%, respectively. Moreover, Leu and Ala also accounted for a higher proportion in Se-TAPepI-1 and Se-TAPepI-2. Nevertheless, there were still significant variations in the allocation of some amino acids between the two fractions. For instance, Se-TAPepI-2 contained a higher content of Asp, Thr, Ser, and Glu, while Se-TAPepI-1 contained a higher content of Gly and Leu. Notably, the contents of hydrophobic and aromatic amino acids of Se-TAPepI-1 were higher than in Se-TAPepI-2. The antioxidant activity showed a positive correlation with hydrophobic amino acids, including Ala, Pro, Val, Leu, Phe, and Met, attributed to the presence of the imidazole ring as a crucial proton donor [41]. It has been reported that antioxidant peptides can enter target organs smoothly through hydrophobic interaction with the membrane lipid bilayer by virtue of their hydrophobicity, thus showing a great performance in scavenging free radicals [42]. The total hydrophobic amino acid proportion in Se-TAPepI-1 was 46.13%, which may partly contribute to the high radical scavenging activity of mixed peptides. It was reported that the enhancement of aromatic amino acids like Tyr and Phe could significantly increase the antioxidant of peptides [43,44]. Interestingly, some sulfur-containing amino acids, such as Cys (0.15%) and Met (1.01%) in Se-TAPepI-1, might provide protein binding sites for Se [45]. These findings further confirmed that Se plays a positive role in the radical scavenging activities of mixed peptides from Se-containing tea protein hydrolysate.

#### 3.4.4. Amino Acid Sequence Analysis

The amino acid sequences of Se-TAPepI-1 and Se-TAPepI-2 were determined using UPLC-Q-TOF-MS. As shown in Figure 4 and Figure 5, two peptides (>1%) from Se-TAPepI-1 were identified as Leu-Pro-Met-Phe-Gly (LPMFG) and Tyr-Pro-Gln-Ser-Phe-Ile-Arg (YPQSFIR), respectively. Also, three peptides (>0.5%) from Se-TAPepI-2 were identified as Gly-Val-Asn-Val-Pro-Tyr-Lys (GVNVPYK), Lys-Gly-Gly-Pro-Gly-Gly (KGGPGG), and Gly-Asp-Glu-Pro-Pro-Ile-Val-Lys (GDEPPIVK), respectively. It was observed that there was a high percentage of hydrophobic amino acid in five peptide sequences, whose N-terminal or C-terminal was mainly occupied by Gly or Lys. This was primarily since papain is an endopeptidase that cleaves peptide bonds at the N-terminal side of Gly or C-terminal of Lys, resulting in peptides chains containing independent Gly or Lys at the N-terminal or C-terminal [46]. Peptides with Val, Leu, Tyr, and Ala residues at the N-terminus displayed a notable antioxidant activity, and the same held true for peptides with Arg at the C-terminus [47,48,49]. Moreover, in the peptide sequence, it has been demonstrated that the inclusion of hydrophobic amino acids (Leu, Val, and Phe), hydrophilic and basic amino acids (His, Pro, and Lys), as well as aromatic amino acids (Phe and Tyr), significantly contributes to the overall heightened antioxidant activity of the peptides [50]. It was evident that the peptides LPMFG and YPQSFIR identified from Se-TAPepI-1 exhibited the similar structural features described above. Compared with Se-TAPepI-1, the presence of hydrophilic Gly at the end of the peptide chain in Se-TAPepI-2 led to the slight insufficiency in antioxidant ability of the peptides [50].

Many studies have shown that antioxidant peptides derived from plant protein have MWs ranging from 500 Da to 1800 Da, which generally consist of 2 to 20 amino acids [51,52]. Obviously, the low MW of the five identified peptides may contribute to their antioxidant activity. In addition to the MW, it was observed that the five identified sequences consisted of a high percentage of hydrophobic amino acid: 88.19% for LPMFG, 58.21% for YPQSFIR, 57.14% for GVNVPYK, 16.67% for KGGPGG, and 50% for GDEPPIVK. Hydrophobic amino acid residues, such as Phe, Tyr, and Trp residues, could chelate the pro-oxidant metal ions due to the electron-dense aromatic rings, thus resulting in the high antioxidant activity [51,53]. Correspondingly, these typical amino acids were found in the two identified sequences from Se-TAPepI-1 and Se-TAPepI-2, which further corroborated the significant role of these hydrophobic amino acids in the antioxidant activity of tea peptides.

### 3.5. Evaluation of Cell Model for Preventing H_2_O_2_-Induced Peroxidation Damage

A cell model of H_2_O_2_-induced oxidative damage was established to further evaluate the antioxidant activity of Se-TAPepIs. As depicted in Figure 6a, exposure to H_2_O_2_ resulted in a decrease in cell viability for LO2 cells, while coincubation with Se-TAPepIs effectively alleviated the damage of LO2 cells. Furthermore, the H_2_O_2_ treatment triggered a large release of ROS in cells (Figure 6b,c), while the treatment with Se-TAPepI-1 and Se-TAPepI-2 dramatically reduced the formation of ROS, and Se-TAPepI-1 showed a better regulatory effect. The expression of genes and the signaling pathway associated with oxidative stress was explored to investigate the possible mechanism of Se-TAPepIs relieving H_2_O_2_-induced oxidative damage. As shown in Figure 7a–d, compared with the MC group, the co-incubation of Se-TAPepI-1 or Se-TAPepI-2 significantly upregulated the mRNA expression of NRF2 and Hmox1 and reduced the accompanying mRNA expressions of IL-1-β and TNF-α. Moreover, the treatment with Se-TAPepI-1 or Se-TAPepI-2 also reduced the expression of NRF2, which promote the expression of the antioxidant response element (ARE) including HO-1 and NQO1 (Figure 7e,f). As expected, the regulation of Se-TAPepI-1 with the hydrophilic amino acid at the terminus of the peptide chain had more impact than that of Se-TAPepI-2.

The stimulation of H_2_O_2_ can cause the oxidation of protein as well as DNA damage in cells and constantly produces •OH radicals, thus leading to the generation of new H_2_O_2_ and triggering a chain reaction of oxidative stimulation [54]. Under the stimulation of H_2_O_2_, excessive ROS disrupt the cellular defense mechanisms, leading to oxidative stress and resulting in irreversible damage to mitochondria and an impairment of cell structure and function [55,56]. Our results showed that Se-TAPepIs could effectively reduce the increase in intracellular ROS caused by H_2_O_2_ stimulation, which was in line with the finding of Wu et al. [54]. NRF2 is a vital transcription factor that is responsive to oxidants and holds a critical function in governing the cellular response to oxidative stress through defense mechanisms [57]. The activation of NRF2 can regulate the transcription of ARE with cytoprotection and detoxification effects, such as HO-1 and NQO1, to maintain cell redox homeostasis and reduce cell oxidative stress [58]. The catalytic subunit of GCLC is an enzyme that plays a pivotal role in initiating and controlling the rate-limiting step of glutathione biosynthesis, and the lack of glutathione is related to the increase in the sensitivity of the host to oxidative stress [59]. Following ROS exposure, NRF2 migrates to the cell nucleus and attaches to the antioxidant response element region, thereby triggering the transcription of target genes and influencing the function of relevant antioxidant enzymes. Excessive ROS not only induce cell vitality reduction and apoptosis but also secrete proinflammatory factors, thus leading to inflammation [60]. Similarly, Se-TAPepIs inhibit the production of ROS by increasing the activity of antioxidant enzymes. Our current results indicate that Se-TAPeps can reduce ROS production and inflammation formation by activating the NRF2/ARE system, further preventing the decrease in cell vitality.

## 4. Conclusions

In this study, two novel antioxidant peptides were successfully prepared and screened from Se-enriched tea protein hydrolysates by hydrolysis with papain. The preliminary cell model analysis of peroxidation damage indicated that Se-TAPepIs could ameliorate H_2_O_2_-induced ROS release via activating the NRF2/ARE pathway, which might be correlated to their Se content, typical amino acid composition and sequence, and unique structure. These findings suggest that the Se-containing antioxidant peptide from Se-enriched tea could serve as a natural substitute for synthetic antioxidants in dietary supplement for improving human health. Moreover, further research to evaluate the bioactivity of Se-enriched tea protein hydrolysates using an animal model has been performed, which will lay the foundation for the application and popularization of Se-enriched tea antioxidant peptides.

## Figures and Tables

**Figure 1 foods-12-04105-f001:**
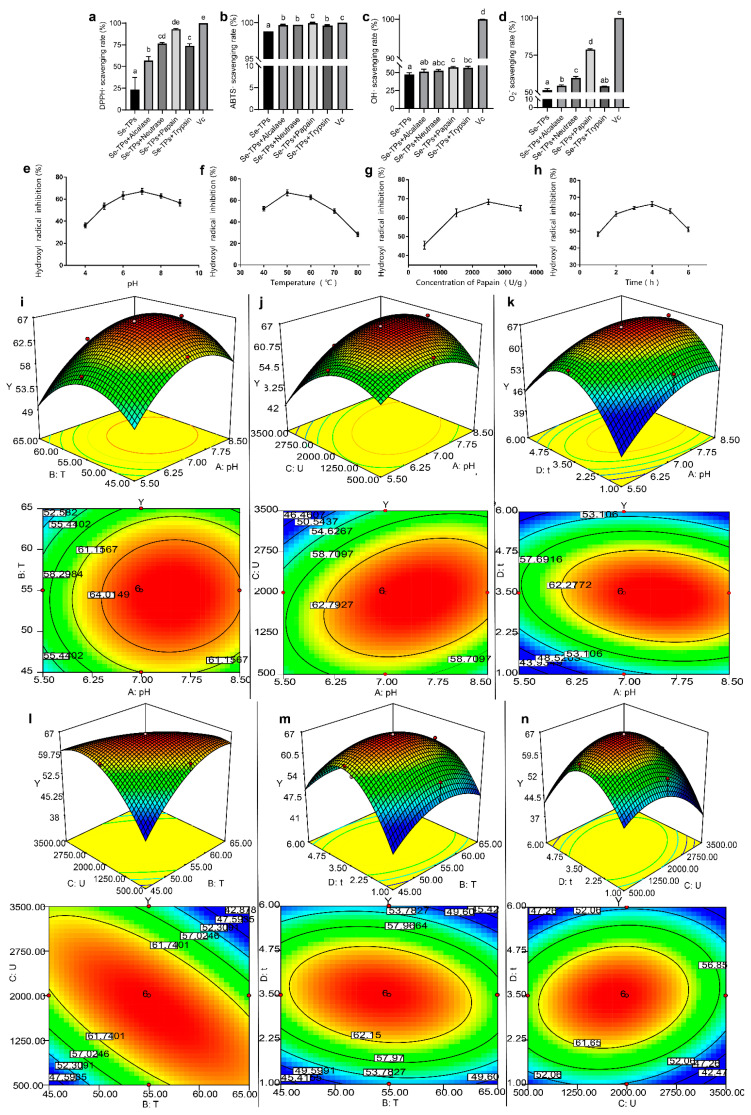
Effects of different hydrolases on the free radical scavenging activities of Se-enriched alkali-soluble tea protein hydrolysate, including DPPH• (**a**), ABTS• (**b**), •OH (**c**), and O_2_^−^ (**d**). Effects of pH (**e**), hydrolysis temperature (**f**), enzyme concentration (**g**), and hydrolysis time (**h**) on the •OH scavenging. Response surface plots for (**i**) Y = f (B, A), (**j**) Y = f (C, A), (**k**) Y = f (D, A), (**l**) Y = f (C, B), (**m**) Y = f (D, B), and (**n**) Y = f (D, C). Vc is vitamin C and mean values with different letters (a, b …) were significantly different in experiment (*p* < 0.05).

**Figure 2 foods-12-04105-f002:**
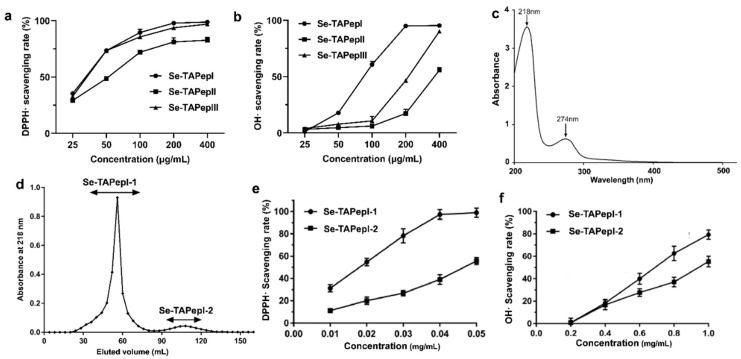
Effects of ultrafiltration components Se-TAPepI (<1 kda), Se-TAPepII (1–10 kda), Se-TAPep3III (10–100 kda) on the scavenging activities of DPPH• (**a**) and •OH (**b**). The UV scanning analysis of Se-TAPepI (**c**). The gel elution curve of Se-TAPepI (**d**). Effects of Se-TAPepI-1 and Se-TAPepI-2 on the scavenging activities of DPPH• (**e**) and •OH (**f**).

**Figure 3 foods-12-04105-f003:**
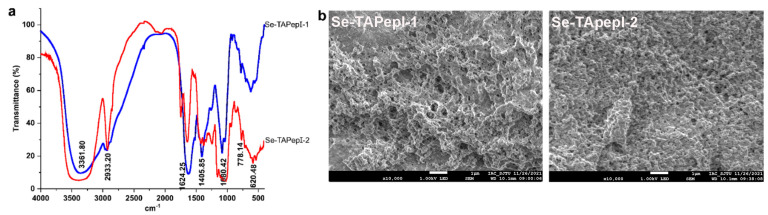
Fourier transform infrared spectra analysis of Se-TAPepI-1 and Se-TAPepI-2 (**a**). Scanning electron microscope of Se-TAPepI-1 and Se-TAPepI-2; the scale is marked in the figure (scale bar = 1 μm) (**b**).

**Figure 4 foods-12-04105-f004:**
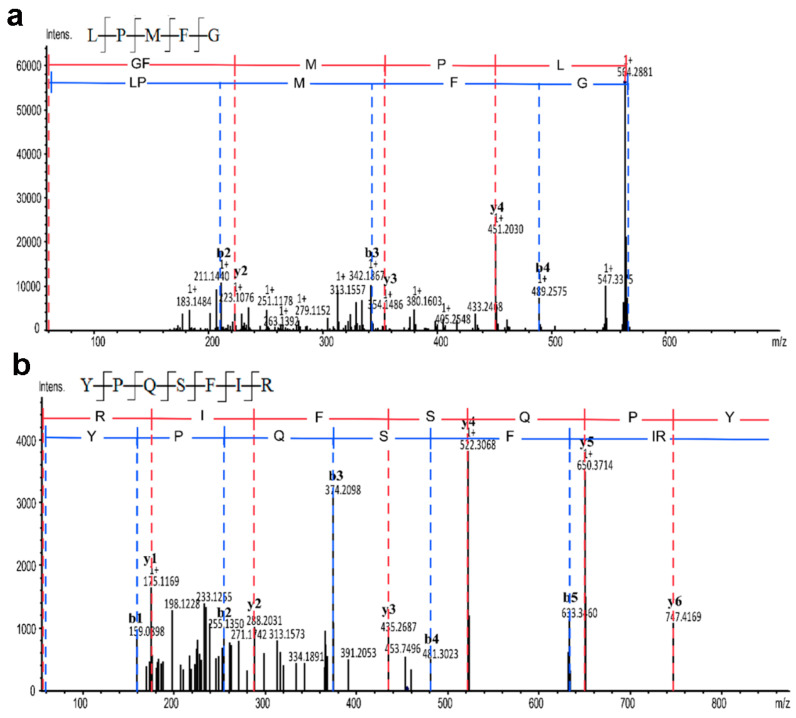
Mass spectrum analysis of the antioxidant peptide derived from Se-TAPepI-1. Mass spectrum of GVNVPYK (**a**); mass spectrum of KGGPHG (**b**), the red and blue lines represent the two fragmented forms of secondary mass spectrometry of the target peptide chain.

**Figure 5 foods-12-04105-f005:**
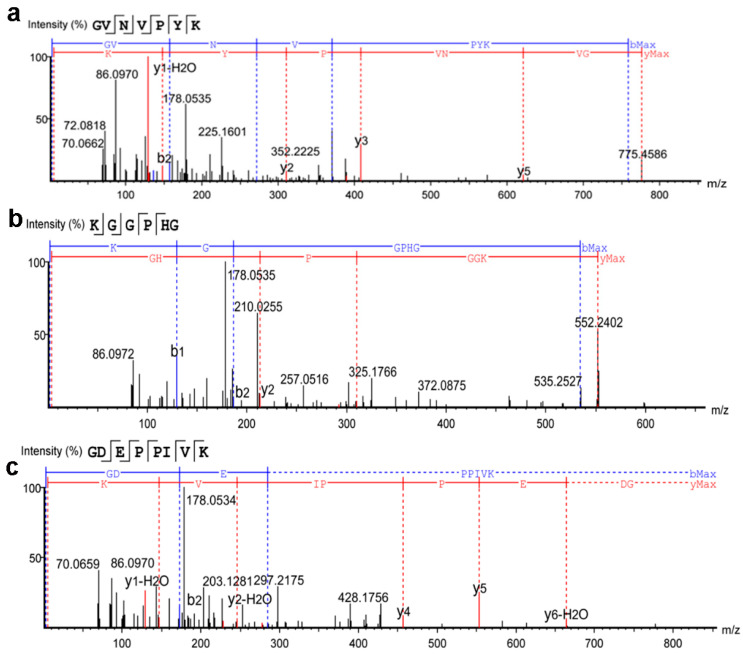
Mass spectrum analysis of the antioxidant peptide derived from Se-TAPepI-2. Mass spectrum of LPMFG (**a**); mass spectrum of YPQSFIR (**b**); mass spectrum of GDEPPIVK (**c**), the red and blue lines represent the two fragmented forms of secondary mass spectrometry of the target peptide chain.

**Figure 6 foods-12-04105-f006:**
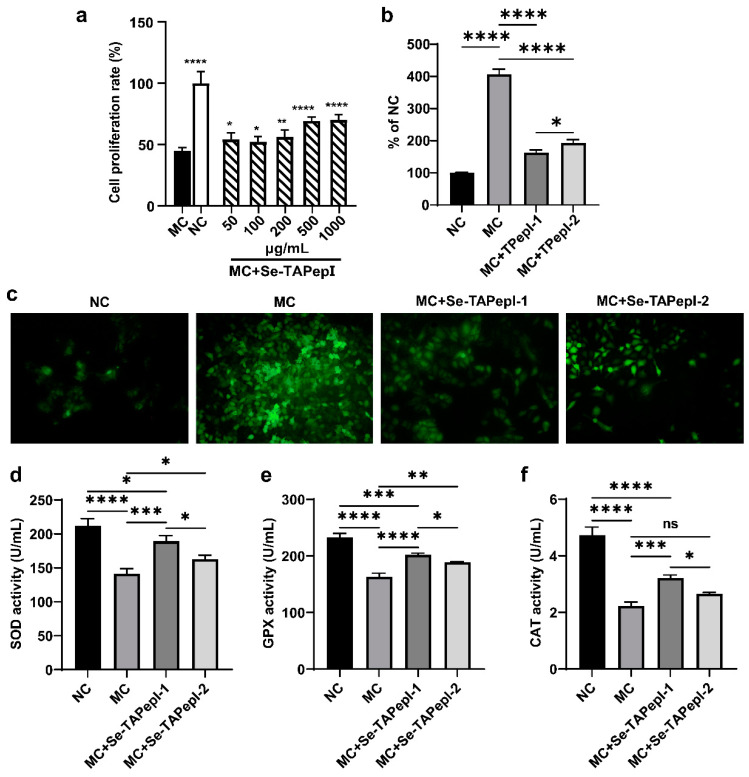
Protection of Se-TAPepIs in H_2_O_2_-induced oxidatively injured LO2 cells. The effect of Se-TAPepIs on cell proliferation in H_2_O_2_-induced oxidatively injured LO2 cells (**a**). The effect of Se-TAPepI-1 and Se-TAPepI-2 on ROS production in H_2_O_2_-induced oxidatively injured LO2 cells (**b**). Images of ROS production observed by fluorescence microscopy (200×) (**c**). The enzyme activity of SOD (**d**), GSH-Px (**e**), and CAT (**f**). ns indicates *p* > 0.05, * *p* < 0.05, ** *p* < 0.01, *** *p* < 0.001, and **** *p* < 0.0001 were compared with the MC. MC (model control) indicates LO2 cells with oxidative damage, and NC (normal control) refers to untreated LO2 cells.

**Figure 7 foods-12-04105-f007:**
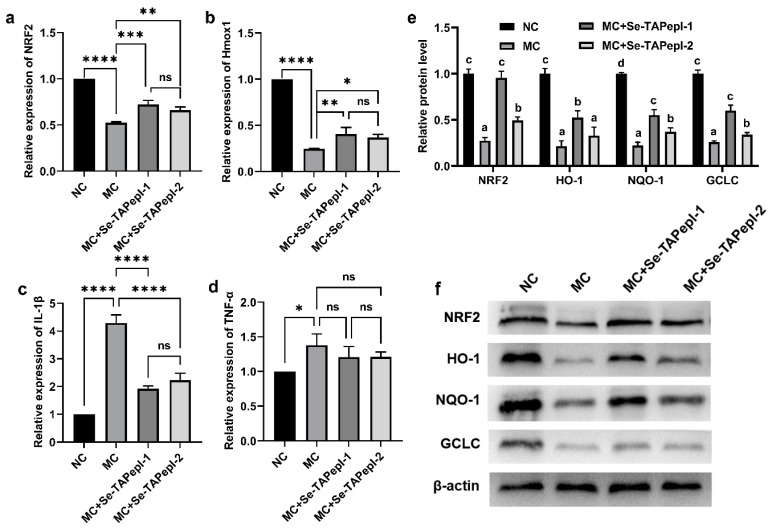
The effects of Se-TAPepIs on mRNA expression and protein levels in H_2_O_2_-induced oxidatively injured LO2 cells. The mRNA expression of NRF2 (**a**), HO-1 (**b**), IL-1β (**c**), TNF-α (**d**) in each treated group of LO2 cells. The protein levels of NRF2, HO-1, NQO-1 and GCLC (**e**) and its grayscale images (**f**) in each treated group of LO2 cells. ns indicates *p* > 0.05, * *p* < 0.05, ** *p* < 0.01, *** *p* < 0.001, and **** *p* < 0.0001 were compared with the MC. MC (model control) indicates LO2 cells with oxidative damage, and NC (normal control) refers to untreated LO2 cells and mean values with different letters (a, b …) were significantly different in experiment (*p* < 0.05).

**Table 1 foods-12-04105-t001:** Amino acid content distribution of Se-TAPeps (nmol%).

Type	Se-TAPepI-1	Se-TAPepI-2
Asp	5.38	9.14
Thr	5.03	7.88
Ser	3.47	11.52
Glu	0.34	10.36
Gly	28.65	12.31
Ala	10.16	10.13
Cys	0.15	0
Val	5.00	7.08
Met	1.01	0
Ile	5.21	3.04
Leu	15.59	8.03
Tyr	5.33	4.19
Phe	6.04	4.89
Lys	6.09	7.33
His	0.84	2.56
Arg	1.71	1.55
Hydrophobic amino acids	43.01	33.17
Aromatic amino acids	11.37	9.08

## Data Availability

The data presented in this study are available on request from the corresponding author.

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
