# Peer review of "Preparation, Characterization, and Antioxidant Properties of Selenium-Enriched Tea Peptides"

_foods, 2023, doi:10.3390/foods12224105_

Round 1
Reviewer 1 Report
Comments and Suggestions for Authors
This study describes the physicochemical properties, oligopeptide sequence, and potential antioxidant mechanisms of antioxidant Se-containing peptides isolated and prepared from Se-enriched tea protein hydrolysate. It is an interesting paper, but there are some unclear points.
1. The authors should explain why the free radical scavenging activities of the hydrolysates obtained by the four proteases differ for each of the four free radicals.
2. Do the peptide fractions with different molecular weights, Se-TAPep I, II, and III, have a common amino acid sequence since they all have antioxidant activity?
3. Among the three peptide fractions with different molecular weights, please explain why the 1-10kDa fraction has the lowest free radical scavenging activity.
4. Please indicate the content of Se in the three peptide fractions and Se-TAPepI-1 and 2. The relationship between Se content and antioxidant activity should be explained.
Author Response
Reviewer 1:
This study describes the physicochemical properties, oligopeptide sequence, and potential antioxidant mechanisms of antioxidant Se-containing peptides isolated and prepared from Se-enriched tea protein hydrolysate. It is an interesting paper, but there are some unclear points.
- The authors should explain why the free radical scavenging activities of the hydrolysates obtained by the four proteases differ for each of the four free radicals.
Response: Thanks for your careful work and comments. Since different enzymes correspond to distinct protease cleavage sites, resulting in varying peptide molecular weights. Additionally, the terminal amino acids of peptide chains differ, thus leading to differing antioxidant properties and free radical elimination rates. In terms of free radicals, the principles governing the elimination of various free radicals by antioxidant compounds vary, for instance, the elimination mechanisms of •O2- involve processes such as electron transfer, hydrogen atom transfer, or reactions with certain aromatic compounds, while the removal of DPPH• occurs in two distinct forms: rapid and slow hydrogen atom transfer. Consequently, even for the same substance, the elimination rates may differ when dealing with different free radicals [1,2]. The revised content was added into the line 250-252 and marked in red.
- Do the peptide fractions with different molecular weights, Se-TAPep I, II, and III, have a common amino acid sequence since they all have antioxidant activity?
Response: Thanks for your careful work and comments. As shown in line 364-370, the antioxidant properties of peptide chains often relate to whether they contain hydrophobic amino acids at their termini. Hence, peptides exhibiting enhanced antioxidant functionality may be attributed to the presence of specific amino acid residues such as Val, Leu, Tyr, and Ala at the N-terminal, or Arg at the C-terminal. Consequently, owing to variations in enzymatic cleavage sites, peptide sequences with different molecular weights like Se-TAPep I, II, and III may share common amino acids, but their respective contributions to antioxidant capacity may be relatively minor. Further “common amino acid sequence” could be explored in the next study.
- Among the three peptide fractions with different molecular weights, please explain why the 1-10kDa fraction has the lowest free radical scavenging activity.
Response: Thanks for your careful work and comments. As mentioned in the text, both the molecular weight and selenium content of peptides can impact the free radical scavenging rate of peptides. Consequently, the lowest molecular weight Se-TAPep I and the highest selenium content Se-TAPep III may be responsible for their higher free radical scavenging rates, while the Se-TAPep II fragments with intermediate levels of molecular weight (1-10 kDa) and selenium content displayed comparatively weaker antioxidant properties. In the future, we will conduct a more in-depth analysis of other potential factors.
- Please indicate the content of Se in the three peptide fractions and Se-TAPepI-1 and 2. The relationship between Se content and antioxidant activity should be explained.
Response: Thanks for your careful work and suggestions. The selenium content in Se-TAPepI-1 and 2 was 2.4 and 0.8 mg/kg, respectively, revealing that for small-molecule peptides like these, selenium content is directly proportional to antioxidant activity. The related contents were added in the line 341-343 and marked in red.
Reviewer 2 Report
Comments and Suggestions for Authors
I revised the manuscript “Preparation, characterization, and antioxidant properties of selenium-enriched tea peptides”. The manuscript brings promising results of peptides hydrolyzed with papain and aspects of their structure and antioxidant function. The results also evaluate the mechanism of action in cells, which confirms the in vitro data and brings more physiological aspects. The study is very well designed and clearly explained in the results section. The data is promising for application of hydrolysates in biofunctional foods. The biggest limitation of the work is the lack of data on the resistance of peptide functionality after the digestive process. It is considered that a peptide will be bioactive if it is capable of remaining intact and bioavailable to reach target tissue. Perhaps this aspect should be discussed as a limitation of the study. Some suggestions are made below:
Methods: It is not clear to me what methodology it was used to evaluate selenium values.
SOD, GPX and CAT activity are note described in methos section.
Line 134: twice DPPH
Line 183: specify “KBr”
Table S1. Add unit of enzyme content
Figure 1: Add “Vc” abbreviation specification in figure 1A, B, C and D.
Line 379-381: It is not clear why the bigger porous structure could bring greater antioxidant activity, what is the relationship between porosity and bioactivity? Bring some reference.
Figure 6, 7: specify MC and NC in the legend of figure 6
Author Response
Reviewer 2:
I revised the manuscript “Preparation, characterization, and antioxidant properties of selenium-enriched tea peptides”. The manuscript brings promising results of peptides hydrolyzed with papain and aspects of their structure and antioxidant function. The results also evaluate the mechanism of action in cells, which confirms the in vitro data and brings more physiological aspects. The study is very well designed and clearly explained in the results section. The data is promising for application of hydrolysates in biofunctional foods. The biggest limitation of the work is the lack of data on the resistance of peptide functionality after the digestive process. It is considered that a peptide will be bioactive if it is capable of remaining intact and bioavailable to reach target tissue. Perhaps this aspect should be discussed as a limitation of the study. Some suggestions are made below:
1.Methods: It is not clear to me what methodology it was used to evaluate selenium values.
Response: Thanks for your careful work and suggestions. The method for determining selenium content were mentioned in the line 162.
2.SOD, GPX and CAT activity are note described in methos section.
Response: Thanks for your careful work and suggestions. The methods for measuring SOD, GPX, and CAT activity were mentioned in the line 208-209.
3.Line 134: twice DPPH
Response: Thanks for your careful work and suggestions. The content has been revised and marked in red.
4.Line 183: specify “KBr”
Response: Thanks for your careful work. The “KBr” is potassium bromide and the information was added in line the 166 and marked in red.
5.Table S1. Add unit of enzyme content
Response: Thanks for your careful work and suggestions. The unit of enzyme content has been added in Table S1, and the revised part was marked in red.
6.Figure 1: Add “Vc” abbreviation specification in figure 1A, B, C and D.
Response: Thanks for your careful work and suggestions. The “VC” abbreviation has been illustrated in the legend of Figure 1.
7.Line 379-381: It is not clear why the bigger porous structure could bring greater antioxidant activity, what is the relationship between porosity and bioactivity? Bring some reference.
Response: Thanks for your careful work and suggestions. As you mentioned, the porous structure is the result of selenization, which enhances the water solubility of tea peptides. There is no clear cause-and-effect relationship for increasing antioxidant properties, so modifications have been made in this regard and the revised part was marked in red. Thank you again for your comment.
8.Figure 6, 7: specify MC and NC in the legend of figure 6
Response: Thanks for your careful work and suggestions. MC (Model control) indicates LO2 cells with oxidative damage, and NC (Normal control) refers to untreated LO2 cells. The information has been added in the legend of Figure 6.
Reviewer 3 Report
Comments and Suggestions for Authors
Dear authors, you will find my comments and suggestions:
COMMENTS AND SUGGESTIONS FOR AUTHORS
Abstract
Line 22. Peptide sequence, and
1. Introduction
Line 39. Change mental diseases to mental disorders or mental health conditions
Line 39. Neurodegenerative diseases
2. Material and methods
Include information of the centrifuge used (model, company, rotor…)
Lines 102 to 105. Rewrite. The words precipitation, precipitate, precipitated are repeated
Line 105. Fnally should be Finally
Line 110 and 256. Neatrasa must be neutrasa. Check all the text and change when necessary
Line 112-113. Why did you hydrolyzed for 4 hour if the hydrolases used are different?
Line 131. Methodology is correctly described, but it is not clear to me why the authors used reference 23 related to antioxidant activity of polysaccharydes extracted from pine cones
Line 150. Change follow to following
Line 151. Include reference for ABTS method
Line 154. The phrase “was evaluated modified by a reported method…” doesn’t sound correct. I suppose it should be: was evaluated by a modified reported method
Line 165. Since the molecular formula C6H6O3 (use subscript) may refer to several compounds, I suggest to include the name of the compound used
Subsection 2.6 and 2.7. Include references to the method used
Line 199. Was used to analysis must be changed to “used for the análisis” or “to analyze”
Subsection 2.10. and 2.10.11. Include reference to the method used
Line 215. Shaker must be shaken
Line 266 and 281. Excellent is an adjective with little or no quantifiable meaning in research, please replace it
Line 290. “Equation” has been written before as equation (lines 140, 158 and 167)
Line 283. Check the position of the radical
Line 292. Write properly the coding variables in the equation
Line 309 and 310. Maybe it will be more correct to use hydrolysis temperature and hydrolysis time instead hydrolytic temperature and hydrolytic time
Lines 332-336. Is it possible that the oxidation state of selenium in the peptide influences the antioxidant activity? If so, discuss that possibility
Line 384. Compares
Line 421. (Leu, Val, and Phe) (His, Pro, and Lys)
Line 428. (a), mass
Line 443. (a), mass…(b), mass
Line 489. Leading to inflammation
4. Conclusion
Lines 496 to 498 are results.
5. References
Line 523. References
It is strongly recommended to cite recent publications (within the last 5-10 years). In this manuscript
13 of 58 references have been published between 11 and 50 years ago (references 1, 13, 19, 20, 23,
29, 42, 43, 44, 45, 46, 50 and 51). Are they so relevant that they cannot be replaced by more recent
ones? I understand that perhaps some of them should stay due to lack of information but, in my
opinión, the majority could be replaced
Please check if some of the following papers could be useful for your manuscript.
Functional properties and structural profiles of water-insoluble proteins from three types of tea
residues. LWT Volume 110, August 2019, Pages 324-331.
https://doi.org/10.1016/j.lwt.2019.04.101
Tea (Camellia sinensis (L.) Kuntze) as an emerging source of protein and bioactive peptides: A
narrative review. https://doi.org/10.1016/j.foodchem.2023.136783. Food Chemistry Volume
428, 1 December 2023, 136783
The Role of Selenium in Health and Disease: Emerging and Recurring Trends,
https://doi.org/10.3390/nu12041049. Nutrients 2020, 12(4), 1049;
https://doi.org/10.3390/nu12041049
none
Author Response
Reviewer 3:
Dear authors, you will find my comments and suggestions:
COMMENTS AND SUGGESTIONS FOR AUTHORS
Abstract
Line 22. Peptide sequence, and
Response: Thanks for your careful work and suggestions. The related content has been revised in line the 13 as your suggestion.
Introduction
Line 39. Change mental diseases to mental disorders or mental health conditions
Response: Thanks for your careful work and suggestions. “mental diseases” was changed to “mental disorders”.
Line 39. Neurodegenerative diseases
Response: Thanks for your careful work and suggestions. The “disease” was changed to “diseases”.
Material and methods
Include information of the centrifuge used (model, company, rotor…)
Lines 102 to 105. Rewrite. The words precipitation, precipitate, precipitated are repeated
Response: Thanks for your careful work and suggestions. Since the repeated protein precipitation that occurred during the protein extraction process, it was inevitable that terms related to precipitation are repeated in the method description.
Line 105. Fnally should be Finally
Response: Thanks for your careful work and suggestions. The “Fnally” was revised as “Finally” and the revised part was marked in red.
Line 110 and 256. Neatrasa must be neutrasa. Check all the text and change when necessary
Response: Thanks for your careful work and suggestions. “Neatrasa” has been revised as “neutrasa” in the whole text, the revised part was marked in red.
Line 112-113. Why did you hydrolyzed for 4 hour if the hydrolases used are different?
Response: Thanks for your careful work and suggestions. As these four enzymes do not have a determined optimal hydrolysis temperature and pH, the optimal hydrolysis time for the same enzyme may vary in different studies[3,4]. Therefore, to obtain smaller enzyme hydrolysate molecular weights, a longer time of 4 hours was selected for enzyme hydrolysis with different enzymes.
Line 131. Methodology is correctly described, but it is not clear to me why the authors used reference 23 related to antioxidant activity of polysaccharydes extracted from pine cones
Response: Thanks for your careful work and suggestions. The reference has been inserted incorrectly, the reference 23 was changed to 10 and marked in red.
Line 150. Change follow to following
Response: Thanks for your careful work and suggestions. The related content was changed and the revised part was marked in red.
Line 151. Include reference for ABTS method
Response: Thanks for your careful work and suggestions. The reference has been added in the text and marked in red.
Line 154. The phrase “was evaluated modified by reported method…” doesn’t sound correct. I suppose it should be: was evaluated by a modified reported method
Response: Thanks for your careful work and suggestions. The content was revised as your suggestion and marked in red,
Line 165. Since the molecular formula C6H6O3 (use subscript) may refer to several compounds, I suggest to include the name of the compound used
Response: Thanks for your careful work and suggestions. C6H6O3 was phloroglucinol, and this method was same as the reference, so this part description was omitted.
Subsection 2.6 and 2.7. Include references to the method used
Response: Thanks for your careful work and suggestions. The related references were added in the section 2.6 and 2.7 and marked in red.
Line 199. Was used to analysis must be changed to “used for the análisis” or “to analyze”
Response: Thanks for your careful work and suggestions. the “analysis” should be changed to “analyze” as your suggestion, the revised content was marked in red.
Subsection 2.10. and 2.10.11. Include reference to the method used
Response: Thanks for your careful work and suggestions. The references of the method in 2.10 were added and marked in red in the text.
Line 215. Shaker must be shaken
Response: Thanks for your careful work and suggestions. “Shaker” has been revised to “shaken” and marked in red in the text.
Line 266 and 281. Excellent is an adjective with little or no quantifiable meaning in research, please replace it
Response: Thanks for your careful work and suggestions. “excellent”was not accurate in the text, and has been replaced with “high” and “great” and marked in red.
Line 290. “Equation” has been written before as equation (lines 140, 158 and 167)
Response: Thanks for your careful work and suggestions. “Equation” was replaced with “equation” and marked in red.
Line 283. Check the position of the radical
Response: Thanks for your careful work and suggestions. The position of the radical of OH was revised in the whole text and marked in red.
Line 292. Write properly the coding variables in the equation
Response: Thanks for your careful work and suggestions. X1, X2, X3, and X4 are the coding variables of pH, temperature, enzyme concentration, and enzymatic hydrolysis time, respectively. The information has been added in the text and marked in red
Line 309 and 310. Maybe it will be more correct to use hydrolysis temperature and hydrolysis time instead hydrolytic temperature and hydrolytic time
Response: Thanks for your careful work and suggestions. The “hydrolytic” was replaced with “hydrolysis” as your suggestion.
Lines 332-336. Is it possible that the oxidation state of selenium in the peptide influences the antioxidant activity? If so, discuss that possibility
Response: Thanks for your careful work and suggestions. It is known to all, organic selenium mainly exists in two oxidized states, selenocysteine and selenomethionine, and exhibits antioxidant activity. However, there have been few studies comparing the antioxidant activities of these two forms of organic selenium, so we did not further discuss this in the article.
Line 384. Compares
Response: Thanks for your careful work and suggestions. The “compared” was replaced with “compares” and marked in red.
Line 421. (Leu, Val, and Phe) (His, Pro, and Lys)
Response: Thanks for your careful work and suggestions. (Leu, Val and Phe) and (His, Pro and Lys) were revised to (Leu, Val, and Phe) and (His, Pro, and Lys) as your suggestion and the revised part was marked in red.
Line 428. (a), mass
Response: Thanks for your careful work and suggestions. The “Mass” has been replaced with “mass” as your suggestion and marked in red.
Line 443. (a), mass…(b), mass
Response: Thanks for your careful work and suggestions. The “Mass” has been replaced with “mass” as your suggestion and marked in red.
Line 489. Leading to inflammation
Response: Thanks for your careful work and suggestions. The sentence has been revised as your suggestion and marked in red.
- Conclusion
Lines 496 to 498 are results.
Response: Thanks for your careful work and suggestions. The content of results has been delated as your suggestion.
References
Line 523. References
It is strongly recommended to cite recent publications (within the last 5-10 years). In this manuscript
13 of 58 references have been published between 11 and 50 years ago (references 1, 13, 19, 20, 23,
29, 42, 43, 44, 45, 46, 50 and 51). Are they so relevant that they cannot be replaced by more recent
ones? I understand that perhaps some of them should stay due to lack of information but, in my opinión, the majority could be replaced. Please check if some of the following papers could be useful for your manuscript.
Functional properties and structural profiles of water-insoluble proteins from three types of tea residues. LWT Volume 110, August 2019, Pages 324-331.
https://doi.org/10.1016/j.lwt.2019.04.101
Tea (Camellia sinensis (L.) Kuntze) as an emerging source of protein and bioactive peptides: A narrative review. https://doi.org/10.1016/j.foodchem.2023.136783. Food Chemistry Volume 428, 1 December 2023, 136783
The Role of Selenium in Health and Disease: Emerging and Recurring Trends,
https://doi.org/10.3390/nu12041049. Nutrients 2020, 12(4), 1049;
https://doi.org/10.3390/nu12041049
Response: Thanks for your careful work and suggestions. Your suggested references are useful and have been replaced in the text, and most of the references you marked have been replaced with recent references expect some classical references like references 19 and 20. Thank you again for your such careful work in my article.
Reference
- Arts, M.J.T.J.; Dallinga, J.S.; Voss, H.; Haenen, G.R.M.M.; Bast, A. A Critical Appraisal of the Use of the Antioxidant Capacity ( TEAC ) Assay in Defining Optimal Antioxidant Structures. 2003, 80, 409–414.
- Schaich, K.M.; Tian, X.; Xie, J. Hurdles and Pitfalls in Measuring Antioxidant Efficacy : A Critical Evaluation of ABTS , DPPH , and ORAC Assays. J. Funct. Foods 2015, 14, 111–125, doi:10.1016/j.jff.2015.01.043.
- Zarei, M.; Ebrahimpour, A.; Abdul-Hamid, A.; Anwar, F.; Bakar, F.A.; Philip, R.; Saari, N. Identification and Characterization of Papain-Generated Antioxidant Peptides from Palm Kernel Cake Proteins. Food Res. Int. 2014, 62, 726–734, doi:10.1016/j.foodres.2014.04.041.
- Luo, H.Y.; Wang, B.; Li, Z.R.; Chi, C.F.; Zhang, Q.H.; He, G. yuan Preparation and Evaluation of Antioxidant Peptide from Papain Hydrolysate of Sphyrna Lewini Muscle Protein. Lwt 2013, 51, 281–288, doi:10.1016/j.lwt.2012.10.008.